# Identification of the Pathogen Causing Leaf Spot in *Zinnia elegans* and Its Sensitivity to Five Fungicides

**DOI:** 10.3390/pathogens11121454

**Published:** 2022-12-01

**Authors:** Yu Liu, Qiuyu Yao, Shuang Liang, Cheng Li, Xiangsheng Chen, Zhong Li

**Affiliations:** 1Key Laboratory of Agricultural Microbiology, College of Agriculture, Guizhou University, Guiyang 550025, China; 2Provincial Key Laboratory of Insect Resources Development and Utilization, Guizhou University, Guiyang 550025, China

**Keywords:** *Zinnia elegans* Jacq., leaf spot disease, *Nigrospora musae*, fungicides

## Abstract

*Zinnia elegans* Jacq. is an important, globally cultivated ornamental plant. In August 2021, a leaf spot disease was observed in zinnia in Shibing County, Guizhou, China, with an incidence of approximately 60%. Pathogens were isolated and purified from the infected leaves by tissue isolation, and pathogen strain BRJ2 was confirmed as the pathogen causing the leaf spot. Based on morphology and *ITS*, *TEF-1α*, and *TUB2* sequence analyses, the pathogen was identified as *Nigrospora musae* (McLennan and Hoëtte). The mycelial growth rate method was used to determine the in vitro toxicity of five fungicides to the pathogen. The results showed that 10% difenoconazole provided the strongest inhibitory effect on *N. musae*, with a concentration for 50% of maximal effect (EC_50_) of 0.0658 mg/L; 75% trifloxystrobin·tebuconazole had the second greatest effect, with an EC_50_ of 0.1802 mg/L. This study provides the first report that *N. musae* caused leaf spot disease in *Z. elegans* and provides important guidance for the effective prevention and control of this disease in Guizhou.

## 1. Introduction

Zinnia (*Zinnia elegans* Jacq.) belongs to the Compositae (also known as Asteraceae) family and is an annual herbaceous flower [1] with single, opposite, curled, wrinkled leaves [2] and various floral colors, such as yellow, white, red, orange, purple, and pink [3]. It exhibits a long and prolific flowering period, high adaptability, rapid growth, and easy cultivation [4]. Furthermore, it originated in Mexico, where it is planted throughout the year and is a commonly used ornamental flower for landscaping [5,6]. Zinnia is not only an ornamental plant but also an important medicinal herb in China. The whole plant is used as a medicine to clear heat, dampness, and toxins and has other medicinal value (http://db.kib.ac.cn/CNFlora/HierarchicalSearch.aspx accessed on 22 October 2022). However, many diseases impede the cultivation of zinnia and seriously affect its ornamental value. In zinnia, *Colletotrichum siamense* infection causes anthracnose [7]; *Golovinomyces cichoracearum*, *Podosphaera xanthii*, and *Erysiphe cichoracearum* cause powdery mildew [8,9,10]; and *Stemphylium lycopersici* and *Xanthomonas campestris* pv. cause leaf spot disease [11,12]. Presently, zinnia leaf spot disease and related harmful symptoms have occurred in China and abroad; however, the identification and control of the disease pathogens remain understudied.

The disease can occur in the seedling and adult stages, resulting in round, polygonal, or irregular gray or dark brown lesions accompanied by a yellow halo. The onset of multiple lesions gradually develops into a large necrotic spot, and the lesion becomes grayish white in the center. The disease causes the leaves of zinnia to fall off early and plants to grow slowly, thereby seriously affecting the size of the flowers, shortening the flowering period, and reducing the ornamental value of affected plants.

In August 2021, the incidence of leaf spot disease in zinnia reached approximately 60% in Shibing, Guizhou, China. To identify the pathogenic species causing this disease and determine effective fungicides for its control, samples of infected plants were collected from Shibing, and morphological and molecular biological methods were used to identify the isolated strains. Pathogenicity tests showed that *Nigrospora musae* (McLennan and Hoëtte) caused the leaf spot. To our knowledge, this is the first study on leaf spots caused by *N. musae* in zinnia. In addition, the fungicidal activities of five fungicides against this pathogen were determined via a mycelial growth rate method to provide important guidance for the effective prevention and control of the zinnia leaf spot in Guizhou province.

## 2. Materials and Methods

### 2.1. Sample Collection and Pathogen Isolation and Purification

In August 2021, one leaf each from three *Z. elegans* plants with brown spots were collected from Shibing (27°3′ N, 108°12′ E). The infected leaf tissue was cut into 5 mm × 5 mm sections, soaked in 75% alcohol for 30 s, and washed three times with sterile distilled water. Subsequently, the tissue was dried on sterile paper and placed in a potato dextrose agar (PDA) plate under sterile conditions. After 1–2 d of incubation in the dark at 28 °C, a small amount of mycelium from the edge of each colony was transferred to a new PDA plate for further culturing. Subsequently, an adequate amount of pure culture was mixed with 30% glycerol at a ratio of 1:1, transferred into a freezer tube, and stored at −80 °C.

### 2.2. Pathogenicity Assay

Koch’s postulates were validated by inoculating zinnia leaves using a conidial suspension. All isolates were tested for pathogenicity. All isolated fungal strains were inoculated onto PDA plates and cultured at 25 °C for 7 d. The colonies were rinsed with sterile water to collect the conidia. The spore concentration was determined using a hemocytometer and adjusted to 1 × 10^6^ conidia/mL using sterile water. Subsequently, 100 μL of this suspension was dropped onto each leaf of the healthy zinnia and spread evenly over the leaf surface using a spreading rod. The control leaves were inoculated with an equal amount of sterile water. After inoculation, the plants were placed in a greenhouse at 28 °C with a relative humidity of 85% and a light/dark cycle of 12/12 h, and the incidence of disease was observed and recorded regularly. Each treatment was repeated three times, and the pathogenicity test was repeated twice. After the inoculated plants became infected, the pathogens were isolated and identified again.

### 2.3. Pathogen Identification and Phylogenetic Analyses

The purified strains were inoculated on a PDA medium and cultured at 25 °C for approximately 7 d, during which their color, shape, texture, and other characteristics were observed. Conidial shape and size were observed and recorded using a microscope (Carl Zeiss AG Axioscope 5, Oberkochen, Germany).

The genomic DNA of the pathogen was extracted using a DNA extraction kit (Beijing Tiangen Biochemical Technology Co., Ltd., Beijing, China). Polymerase chain reaction (PCR) was performed to amplify the internal transcribed spacer (*ITS*) regions, beta-tubulin gene (*TUB2*), and translation elongation factor 1-alpha encoding gene (*TEF-1α*) of the extracted DNA using the primer sets ITS1/ITS4 [13,14], EF1-728F/EF1-986R [15], and T1/Bt2b [16,17], respectively (Table 1). PCR was conducted at a final volume of 25 μL, containing 2× TaqMasterMix (12.5 µL) (Shanghai Sangon Biotech Co., Ltd., Shanghai, China), a 10 mol/L solution of each primer (1 µL), template DNA (1 µL), and double-distilled H_2_O (9.5 µL). The amplification conditions were as follows: initial denaturation at 95 °C for 5 min, followed by 35 cycles of denaturation at 94 °C for 30 s, annealing for 30 s (at 52 °C for *ITS*, 60 °C for *TEF-1α*, and 55 °C for *TUB2*), and extension at 72 °C for 30 s, followed by a final supplementary extension at 72 °C for 7 min [18]. The PCR products were examined using agarose gel electrophoresis and sent to Shanghai Sangon Biotech Co., Ltd. for sequencing. The DNA sequence of the pathogen was compared with other sequences deposited in GenBank using the BLAST program for similarity analysis; subsequently, the sequence data regarding *Nigrospora* spp. were downloaded from GenBank to construct the phylogenetic tree (Table 2) of the *ITS, TEF-1α*, and *TUB2* sequences using the maximum likelihood (ML) and Bayesian inference (BI) methods and the CIPRES Science Gateway website (phylo.org) [19]. *Arthrinium malaysianum* was used as an outgroup.

### 2.4. Determination of the Sensitivity of Strain BRJ2 to Fungicides

The antifungal activity of five fungicides against *N. musae* was determined via a growth rate method [20]. The fungicides were diluted with sterile water and mixed into the PDA medium at different volumes to obtain five concentrations. After fully mixing, the solutions were poured into sterile Petri dishes with a diameter of 9 cm; PDA medium mixed with sterile water was used as a control. Each treatment had three replicates. A 5 mm-diameter mycelial disk of *N. musae* was inoculated on each plate. After incubation at a constant temperature of 25 °C for 6 d, colony diameter under each treatment was measured via the cross method, and the inhibition rate was calculated according to the following formula [20]:inhibitory rate (%) = [(D_control_ − D_treatment_)/(D_control_ − 5)] × 100(1)
where D represents the diameter of the *N. musae* colony. The EC_50_ (concentration for 50% of maximal effect) values and confidence intervals of the fungicides were calculated using the IBM SPSS software (SPSS Inc., Chicago, IL, USA) [21].

## 3. Results

### 3.1. Isolation of Pathogens and Pathogenicity Tests

Leaf spots were observed in zinnia in August 2021. During the early stages of infection, round and polygonal brown lesions with yellow halos appeared on the leaves. During the later stages, the lesions gradually expanded into irregular necrotic lesions, and the center of the lesions became grayish white (Figure 1A,B). Three samples of diseased leaves were collected. From these, 15 fungal isolates with different colony morphologies were obtained, and pathogenicity tests were performed on these isolates. The results showed that only the BRJ2 isolate was pathogenic in zinnia. Five days after artificial inoculation with a conidial suspension of the BRJ2 strain, yellowish brown spots appeared on the leaves of formerly healthy zinnia plants (Figure 1D,E), which were consistent with the field symptoms; leaves inoculated with sterile water or other strains showed no leaf spot symptoms (Figure 1C). Pathogens isolated from the artificially inoculated diseased leaves had morphological characteristics similar to those of the strain BRJ2. Therefore, the pathogenicity tests showed that the strain BRJ2 caused leaf spots in zinnia.

### 3.2. Identification of the Strain BRJ2

The colonies of the BRJ2 isolate were villous and rounded after 5 d on the PDA medium. Colonies were initially white; however, the center gradually became grayish black to black. After 7 d, their diameter reached 90 mm (Figure 2A,B). The conidia were solitary, terminal, spherical or subspherical, black, smooth, and 10.2–15.8 μm in diameter (Figure 2C,D); they were borne on transparent vesicles at the tip of each conidiophore. The conidiogenous cells were aggregated, pale brown, and subglobose to ampulliform, with dimensions of 5.5–13 × 4–8.5 μm. Based on the morphological characteristics of BRJ2, the fungus was preliminarily identified as a species of *Nigrospora* [22].

The sequences of the *ITS* of the ribosomal DNA, *TEF-1α*, and *TUB2* of BRJ2 were amplified by PCR and submitted to GenBank (https://www.ncbi.nlm.nih.gov/genbank/ accessed on 22 October 2022) The accession numbers are OP451019, OP627526, and OP699164, respectively. A phylogenetic tree was constructed by combining the *ITS*, *TUB2*, and *TEF-1α* sequences, and *A. malaysianum* was used as the outgroup. The results (Figure 3) showed that the isolated strain BRJ2 and the type strain *Nigrospora musae* (McLennan and Hoëtte) (CBS 319.34, strain number CBS 106.24) were clustered in the same branch; this was supported by an ML of 100% and a BI of 1.00. Therefore, the BRJ2 isolate was identified as *N. musae* based on morphological characteristics and molecular biological identification.

### 3.3. Determination of the Sensitivity of Strain BRJ2 to Fungicides

The results of the sensitivity testing of the BRJ2 isolate to the five fungicides are shown in Table 3. The correlation coefficients for all five fungicides were above 0.90, indicating a high correlation between fungicide dosage and BRJ2 inhibition. In addition, 10% difenoconazole and 75% trifloxystrobin·tebuconazole had the best inhibitory effects on BRJ2, with EC_50_ values of 0.0658 mg/L and 0.1802 mg/L, respectively. These were followed by 50% prochloraz manganese and 1% cnidium lactone, with EC_50_ values of 1.1809 mg/L and 1.4984 mg/L, respectively. In contrast, shenqinmycin had a relatively small effect on the pathogen, with an EC_50_ value of 4.5066 mg/L.

## 4. Discussion

Zinnia plants are widely cultivated in urban green spaces and parks because of their bright colors and long flowering period. Leaf spot disease was observed in cultivated zinnia plants in Shibing, Guizhou. Fifteen isolates were obtained from diseased zinnia leaves, but only BRJ2 was confirmed to be pathogenic by Koch’s postulates. This study identified the pathogen (BRJ2) that caused the leaf spot disease as *N. musae* using traditional morphological and molecular biological methods. As important ascomycetes, *Nigrospora* spp. are distributed worldwide and have a wide host range [23,24]. *N. musae* is a common member of this genus that can cause many plant diseases, such as fruit diseases in *Musa* × *paradisiaca* [25] and leaf diseases in *Camellia sinensis* [26].

Chemical control provides effective ways of preventing and controlling pests and diseases in agriculture and forestry and has many advantages, such as rapid effects and simple methods. However, there are no studies on the control of leaf spot disease in zinnia. Cui et al. reported that six fungicides (prochloraz, difenoconazole, spargon, propiconazole, difenoconazole thiram, and hymexazol) had inhibitory effects on the growth of black spore fungi. Among these, prochloraz had the best inhibitory effect of up to 98% [27]. The present study determined the in vitro antifungal effects of five low-toxicity and high-efficiency fungicides on *N. musae*. The results showed that all five fungicides had antifungal effects. The 10% difenoconazole water-dispersible granules had the strongest antifungal activity, with an EC_50_ value of 0.0658 mg/L, which was considerably better than the other fungicides. Additionally, 75% trifloxystrobin·tebuconazole-dispersible granules also had a good antifungal effect, with an EC_50_ value of 0.1802 mg/L. Nevertheless, these results differ from those reported by another study [27], which may be related to individual and regional differences in strains.

The main active ingredients of these two fungicides are triazoles. Triazole fungicides inhibit ergosterol biosynthesis by inhibiting cytochrome P450 (CYP 450) activity, thereby inhibiting fungal growth [28]. Triazole fungicides exhibit low toxicity, high efficiency, broad-spectrum activity, low dosages, and easy degradation. They have been widely used to prevent and control various diseases caused by fungi, such as ascomycetes and basidiomycetes [29]. Difenoconazole controls rust, anthracnose, sheath blight, and other fungal diseases adequately [30,31,32]. However, owing to the extensive application of triazole fungicides, many important pathogens have developed resistance in the field [33,34,35]. Therefore, we should consider environmental factors, cost, pesticide residues, and pathogen resistance for fungal control and avoid the long-term use of a single fungicide via the use of biological agents. The main components of natural biological agents do not easily induce resistance and can promote plant growth and rooting [21]. Moreover, most botanical fungicides promote plant disease resistance and other comprehensive effects to reduce plant disease. In the present study, the sensitivity of *N. musae* to cnidium lactone and shenqinmycin was examined. The EC_50_ of cnidium lactone was 1.4984 mg/L, which was higher than that of shenqinmycin, indicating a better fungistatic effect. Therefore, when controlling leaf spot disease, cnidium lactone and difenoconazole can be used alternately to improve pathogen control in the field and delay fungicidal resistance. Nevertheless, this test of in vitro antifungal activity is only for reference, and more detailed field trials are required for verification.

## 5. Conclusions

This study identified the pathogen causing zinnia leaf spot as *N. musae* based on morphological observations, molecular identification, and pathogenicity testing. To the best of our knowledge, this is the first study of zinnia leaf spot disease caused by *N. musae*. In addition, the inhibitory effects of five fungicides on the pathogen were determined, and 10% difenoconazole and 75% trifloxystrobin·tebuconazole had the greatest inhibitory effects on the mycelial growth of the pathogen.

## Figures and Tables

**Figure 1 pathogens-11-01454-f001:**
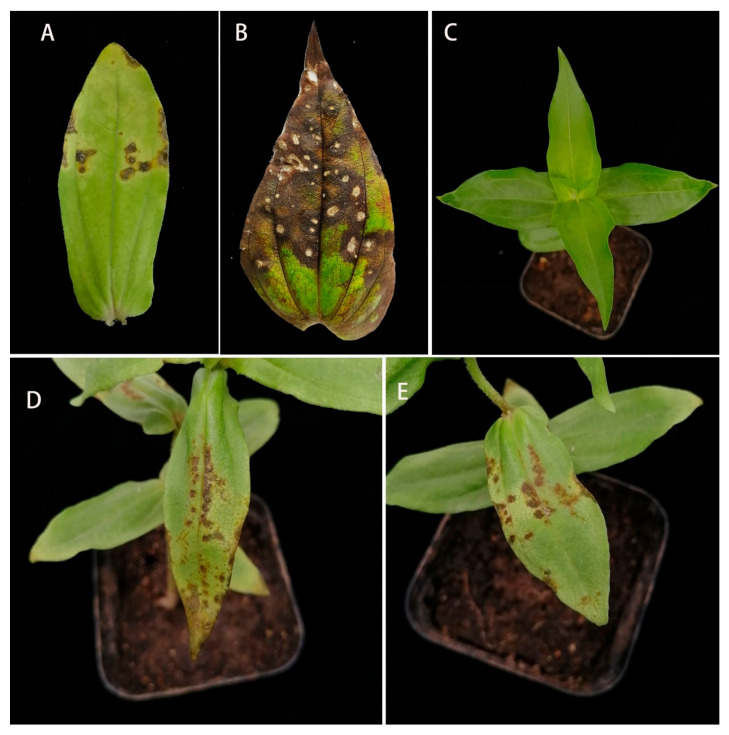
(**A**,**B**) Field symptoms of zinnia leaf spot disease. Healthy plants that were (**C**) inoculated with sterile water and (**D**,**E**) plants presenting symptoms on leaves due to inoculation with BRJ2 after 7 d.

**Figure 2 pathogens-11-01454-f002:**
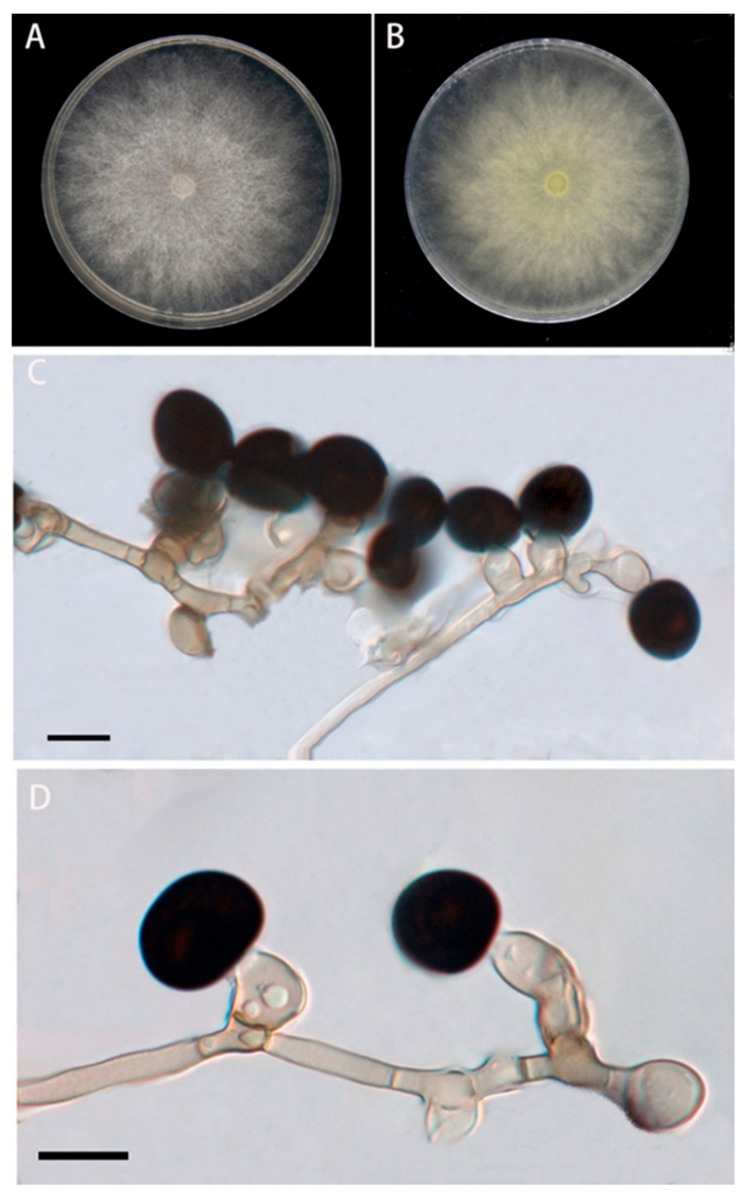
Morphological characteristics of the BRJ2 isolate on PDA after seven days. (**A**) Front of a BRJ2 colony; (**B**) back of a BRJ2 colony. (**C**,**D**) Conidiogenous cells giving rise to conidia. (**C**,**D**) Scale bar = 10 μm.

**Figure 3 pathogens-11-01454-f003:**
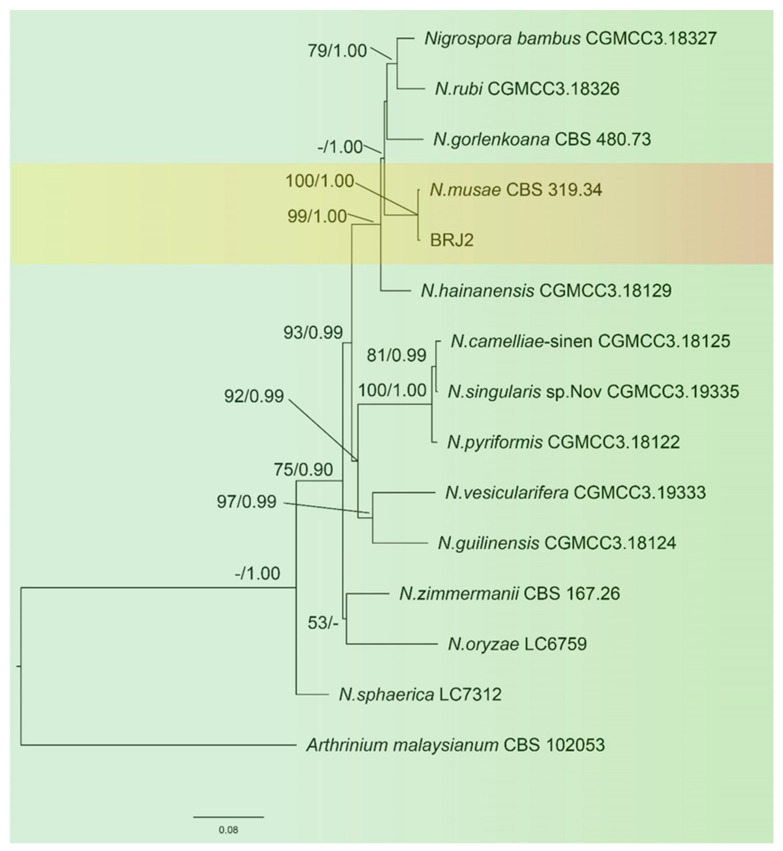
Maximum likelihood (ML) tree of *Nigrospora* spp. based on 13 strains. Maximum likelihood analysis bootstrap support (ML ≥ 50%) and Bayesian posterior probability (PP ≥ 0.90) are shown at the nodes (ML/PP). The tree was rooted using *Arthrinium malaysianum* (CBS 102053). The scale bar indicates 0.08 expected changes per site.

**Table 1 pathogens-11-01454-t001:** PCR primer sequences.

Target Sequence	Primer	Primer Sequence (5′-3′)	Reference
ITS	ITS1	CTTGGTCATTTAGAGGAAGTAA	[13]
	ITS4	TCCTCCGCTTATTGATATGC	[14]
TEF1-α	EF1-728F	CATCGAGAAGTTCGAGAAGG	[15]
	EF1-986R	TACTTGAAGGAACCCTTACC	
TUB2	T1	AACATGCGTGAGATTGTAAGT	[16]
	Bt2b	ACCCTCAGTGTAGTGACCCTTGGC	[17]

**Table 2 pathogens-11-01454-t002:** Sequence information for the strains used for multigene phylogenetic analysis in this study.

Species	Strain No.	GeneBank Accession Number		
		ITS	TUB	TEF1-α
*Nigrospora. bambus*	CGMCC3.18327 *	KY385307	KY385319	KY385313
*Nigrospora. camelliae-sinen*	CGMCC3.18125 *	KX985986	KY019460	KY019293
*Nigrospora. gorlenkoana*	CBS 480.73 *	KX986048	KY019456	KY019420
*Nigrospora. guilinensis*	CGMCC3.18124 *	KX985983	KY019459	KY019292
*Nigrospora. hainanensis*	CGMCC3.18129 *	KX986091	KY019464.1	KY019415
*Nigrospora. musae*	CBS 319.34 *	NR153478	KY019455	KY019419
*Nigrospora. oryzae*	LC6759	KX986054	KY019572.1	KY019374.1
*Nigrospora. pyriformis*	CGMCC3.18122 *	KX985940	KY019457.1	KY019290.1
*Nigrospora. rubi*	CGMCC3.18326 *	KX985948	KY019475	KY019302
*Nigrospora. sphaerica*	LC 7312	KX985935	KY019618	KY019414
*Nigrospora. zimmermanii*	CBS 167.26	KY385308	KY385318	KY385312
*Nigrospora. singularis sp. nov*	CGMCC 3.19335	MN215793	MN329956	MN264032
*Nigrospora. vesicularifera*	CGMCC 3.19333	MN215812	MN329975	MN264051
*Arthrinium malaysianum*	CBS 102053	KF144896.1	KF144988.1	KF145030.1

Note: * = ex-type strains.

**Table 3 pathogens-11-01454-t003:** In vitro toxicity of five fungicides against *N. musae*.

Fungicides	Concentrations (ug/mL)	Regression Equation of Toxicity	EC_50_ (mg/L)	Correlation Coefficient (r)
Prochloraz Manganese 50% WP	0.25, 0.5, 1, 2, 4	y = 1.3000x + 4.9061	1.1809 ± 0.11	0.9884
Difenoconazole 10% WG	0.25, 0.74, 2.22, 6.66, 20	y = 0.5188x + 5.6129	0.0658 ± 0.08	0.959
Trifloxystrobin·Tebuconazole 75% WG	0.0375, 0.075, 0.15, 0.3, 0.6	y = 1.3004x + 5.9678	0.1802 ± 0.07	0.9965
Cnidium Lactone 1% EW	0.375, 0.75, 1.5, 3,9	y = 1.1128x + 4.8046	1.4984 ± 0.16	0.9696
Shenqinmycin 1% SC	1.5625, 3.125, 6.25, 12.5, 25	y = 1.4781x + 4.0336	4.5066 ± 0.16	0.9826

## Data Availability

The datasets generated and/or analyzed during the study are available from the corresponding author upon reasonable request.

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
