# Peer review of "Identification of the Pathogen Causing Leaf Spot in Zinnia elegans and Its Sensitivity to Five Fungicides"

_pathogens, 2022, doi:10.3390/pathogens11121454_

Round 1

Reviewer 1 Report

The authors identify Nigrospora musae as the causal agent on leaf spot on Zinnia. Five fungicides were tested to determine the growth inhibition potential against this single isolate of N. musae.

Concerns:
Fifteen isolates were collected from diseased leaves. Were all isolates tested for pathogenicity?

Only one isolates was pathogenic. How many plants were sampled? If Nigrospora is the pathogen responsible for the leaf spot, the expectation would be to find more isolates?

Title: change six to five

L43-44: Not sure what the authors mean by this phrase, "causes early 43 leaves of zinnia, a plant growth lentor,"
L49: First mention on name, expand genus name
L51: says, "five synthetic fungicides", discrepancy between number of fungicides tested, title says six

L68: "Subsequently, 100 μL of the spore suspension was evenly sprayed onto each leaf of the healthy zinnia." How was 100μL sprayed?

L89: Use β-tubulin (TUB2), so reader knows the abbreviation.

L93: Italicize genus name

L 97: Table 2. Change polygene to multigene
Italicize all species names.
Remove period from Nigrospora.

L100: Change the header, as no natural antifungal agents are being tested in this study, only synthetic fungicides.

Also this header is an exact match to Citation - J Fungi (Basel). 2022 Mar; 8(3): 239.
 "2.4. Antimicrobial Activity of Natural Antifungal Agents on Mycelial Growth"

L104: remove extra period.

L116: In vitro, not Indoor

L133: The phrase "Natural field symptoms of brown leaf spot on kiwifruit."
is an exact match to Citation: Chen, J.; Ran, F.; Shi, J.; Chen, T.; Zhao, Z.; Zhang, Z.; He, L.; Li, W.; Wang, B.; Chen, X.; et al. Identification of the Causal Agent of Brown Leaf Spot on Kiwifruit and Its Sensitivity to Different Active Ingredients of Biological Fungicides. Pathogens 2022, 11, 673. https://doi.org/10.3390/pathogens11060673

L133: kiwifruit??. Change (c) to (C)
L134: Change (C,D) to (D,E)

L151: Change polygene to multigene

L181: Italicize and change to Musa x paradisiaca. And italicize Camellia sinensis

L188: antibacterial??
L190: antibacterial??
L194: antibacterial??
L209: bacteriostatic??
L213: antibacterial??

L192-193: What other studies? Provide citations.

Avoid using the word antibacterial for a fungicide.

L208: Osthole has not been used previously in the manuscript

There are some lines that appear to copied from other manuscripts,

L100: This header "2.4. Antimicrobial Activity of Natural Antifungal Agents on Mycelial Growth" is an exact match to Citation - J Fungi (Basel). 2022 Mar; 8(3): 239.

L133: The phrase "Natural field symptoms of brown leaf spot on kiwifruit."
is an exact match to Citation: Pathogens 2022, 11:673. https://doi.org/10.3390/pathogens11060673

Author Response

Response to Reviewer 1 Comments

I am very grateful to the reviewers for their suggestions. I carefully considered your suggestion, made the following introduction and response and in the manuscript has been modified highlighted in red font. If there are still questions, I will reply as soon as possible

Point 1: Fifteen isolates were collected from diseased leaves. Were all isolates tested for pathogenicity?.

Response 1: Thank for your question All isolates were tested for pathogenicity. Corresponding modifications have been made in the manuscript.

Point 2: Only one isolates was pathogenic. How many plants were sampled? If Nigrospora is the pathogen responsible for the leaf spot, the expectation would be to find more isolates?

Response 2: Thank for your question .Yes, only one isolate is pathogenic. A total of 3 diseased leaves were collected for routine tissue isolation, and 15 strains with different morphologies were obtained. The Koch 's postulates verified that strain BRJ2 could cause leaf disease, and the symptoms were similar to those in the field. Subsequent studies used strain BRJ2 as the target.

Point 3 : L43-44: Not sure what the authors mean by this phrase, "causes early 43 leaves of zinnia, a plant growth lentor,"

Response 3: " Thank for your question. "causes early 43 leaves of zinnia, a plant growth lentor,"mean The disease causes the leaves of Zinnia to fall off early, plants grow slowly. This statement in the manuscript has been modified to make it easier to understand. (in L43-44)

Point 4 : L49: First mention on name, expand genus name

Response 4: Thanks for the reminder. The manuscript has been modified as required. “N. musae” to Nigrospora musae.

Point 5 : L51: says, "five synthetic fungicides", discrepancy between number of fungicides tested, title says six

Response 5: Thank you for your questions. Five fungicides were used in the experiment. The title of the manuscript has been modified.

Point 6 : L68: "Subsequently, 100 μL of the spore suspension was evenly sprayed onto each leaf of the healthy zinnia." How was 100μL sprayed?

Response 6: Thank you for your questions. I have corrected the following information: Subsequently, 100 μL of spore suspension is dropped onto each leaf of the healthy zinnia and spread evenly over the leaf using a spreading rod.

Point 7 : L89: Use β-tubulin (TUB2), so reader knows the abbreviation.

Response 7: Thanks for your valuable advice The manuscript has been modified as required

Point 8 : L93: Italicize genus name

Response 8: Thanks for your valuable advice.The manuscript has been modified as required

Point 9 : L 97: Table 2. Change polygene to multigene. Italicize all species names.
Remove period from Nigrospora.

Response 9: Thank you for your advice. The manuscript has been modified as required.

Point 10 : L100: Change the header, as no natural antifungal agents are being tested in this study, only synthetic fungicides.

Response 10: Thank you for your advice. The manuscript has been modified as required. (Determination of sensitivity of strain BRJ2 to Synthetic fungicides)

Point 11 : L104: remove extra period.

Response 11: Thank you for your advice. The manuscript has been modified as required.

Point 12 : L116: In vitro, not Indoor.

Response 12: Thank you for your advice . The manuscript has been modified as required.

Point 13 : L133: The phrase "Natural field symptoms of brown leaf spot on kiwifruit."
is an exact match to Citation: Chen, J.; Ran, F.; Shi, J.; Chen, T.; Zhao, Z.; Zhang, Z.; He, L.; Li, W.; Wang, B.; Chen, X.; et al. Identification of the Causal Agent of Brown Leaf Spot on Kiwifruit and Its Sensitivity to Different Active Ingredients of Biological Fungicides. Pathogens 2022, 11, 673. https://doi.org/10.3390/pathogens11060673

Response 13: Thank you for your advice. The manuscript has been revised to Field symptoms of zinnia leaf spot disease.

Point 14 : L133: kiwifruit??. Change (c) to (C) L134: Change (C,D) to (D,E)

Response 14: Thank you for your advice .The manuscript has been modified as required.

Point 15 : L151: Change polygene to multigene

Response 15: Thank you for your advice .The manuscript has been modified as required.

Point 16 : L181: Italicize and change to Musa x paradisiaca. And italicize Camellia sinensis

Response 16: Thank you for your advice .The manuscript has been modified as required.

Point 17 : L188: antibacterial?? L190: antibacterial?? L194: antibacterial?? L219: bacteriostatic?? L213: antibacterial??

Response 17: Thank you for your advice. The manuscript has been revised antibacterial to antifungal, and revised bacteriostatic to fungistatic

Point 18 : L192-193: What other studies? Provide citations.

Response 18: Thank you for your question. Other studies refers to the literature cited above [27]. The manuscript has been modified as required.

Point 19 : Avoid using the word antibacterial for a fungicide.

Response 19: Thank you for your valuable suggestions. The manuscript has been revised antibacterial to antifungal

Point 20 : L208: Osthole has not been used previously in the manuscript

Response 20: Thanks for the reminder. I have corrected the information The manuscript has been revised Osthole to cnidium lacton.

Point 21: There are some lines that appear to copied from other manuscripts,

L100: This header "2.4. Antimicrobial Activity of Natural Antifungal Agents on Mycelial Growth" is an exact match to Citation - J Fungi (Basel). 2022 Mar; 8(3): 239.

L133: The phrase "Natural field symptoms of brown leaf spot on kiwifruit."
is an exact match to Citation: Pathogens 2022, 11:673. https://doi.org/10.3390/pathogens11060673

Response 21: The manuscript has referred to these two articles in the writing process. I have corrected the information.

Reviewer 2 Report

Dear Authors, 

The study on the "Identification of the Pathogen of Leaf Spot of Zinnia and Its Sensitivity to six synthetic fungicides" interests the work you have completed. 

I have annotated my several concerns on the body of MS, and the file is attached herewith. I request you to complete the revisions. A few of my concerns are as below. 

1.  line 60: "After 1–2 dark incubations at 28 °C, a small" explain 1-2 are days?

2. Line 82: elaborate ITS and TEF1 at its first mention, TEF1 should be in italics.

3. Line 94. In a phylogenetic tree which out-group was considered for re-rooting the tree?

4.   Table 1 Pls give references to all the barcodes. 

5. Table 2. the scientific names should be in italics 

6. Line 110 Inhibitory rate (%), pls give a citation for the formula you used for these calculations. 

7. 3.2. Identification of Strain BRJ2: Pls provide the detailed morphology l including conidial measurements, conidiogenesis, details on conidiophores other morphological details too. Has this strain been deposited to the national/international microbial repository (IDA)? if not, pls submit and provide the accession number.

8. Line 152, give the name of the authors too. Once identified at first mention, it should be written as Nigrospora musae McLennan & Hoëtte.

9.  Line 150 Pls provide the link to genebank 

Kindly incorporate the above changes in the MS and resubmit for further decisions. 

Regards, 

Author Response

Response to Reviewer 2 Comments

I am very grateful to the reviewers for their suggestions. I carefully considered your suggestion, made the following introduction and response and in the manuscript has been modified highlighted in red font. If there are still questions, I will reply as soon as possible

Point 1:. line 60: "After 1–2 dark incubations at 28 °C, a small" explain 1-2 are days?

Response 1: Thank you for your question. The manuscript has been modified to “After 1–2 day dark incubations at 28 °C”

Point 2. Line 82: elaborate ITS and TEF1 at its first mention, TEF1 should be in italics.

Response 2: Thank you for your advice. The manuscript has been modified as required. (Polymerase chain reaction (PCR) amplified the internal transcribed spacer(ITS) region, beta-tubulin (TUB2), and translation elongation factor 1-alpha encoding gene (TEF-1α) with the primer (Table 1) sets ITS1/ITS4[13,14] and EF1-728F/EF1-986R[15] T1/Bt2b[16,17].)

Point 3. Line 94. In a phylogenetic tree which out-group was considered for re-rooting the tree?

Response 3: Thank you for your question. Yes, out-group mean the roots of the tree.

Point 4. Table 1 Pls give references to all the barcodes.

Response 4: Thank you for your advice. References added in table 1.

Point 5. Table 2. the scientific names should be in italics

Response 5: Thank you for your advice. The manuscript has been modified as required

Point 6. Line 110 Inhibitory rate (%), pls give a citation for the formula you used for these calculations.

Response 6: Thank you for your advice.This inhibitory rate (%) citation that reference [20] has been added to the manuscript.

Point 7. 3.2. Identification of Strain BRJ2: Pls provide the detailed morphology l including conidial measurements, conidiogenesis, details on conidiophores other morphological details too. Has this strain been deposited to the national/international microbial repository (IDA)? if not, pls submit and provide the accession number.

Response 7: Thank you for your advice. The morphological details added in the manuscript is described as follows: Conidiogenous cells aggregated, pale brown, subglobose to ampulliform, 5.5–13 × 4–8.5 μm. Hyaline vesicles delimiting the conidia from conidiogenous cells. Cnidia borne on a hyaline vesicle at the tip of each conidiophore. The strain is not stored in the national/international microbial repository ( IDA ). The accession number is provided in lines 158-159 . The accession numbers were OP451019(ITS), OP627526(TEF-1α), and OP699164(TUB2).

Point 8. Line 152, give the name of the authors too. Once identified at first mention, it should be written as Nigrospora musae McLennan & Hoëtte.

Response 8: Thank you for your advice. The manuscript has been modified as required

Point 9. Line 150 Pls provide the link to genebank?.

Response 9: Thank you for your advice. The manuscript has been modified as required. (GenBank: (https://www.ncbi.nlm.nih.gov/genbank/))